# Arbitrary-Order Finite-Time Corrections for the Kramers–Moyal Operator

**DOI:** 10.3390/e23050517

**Published:** 2021-04-24

**Authors:** Leonardo Rydin Gorjão, Dirk Witthaut, Klaus Lehnertz, Pedro G. Lind

**Affiliations:** 1Forschungszentrum Jülich, Institute for Energy and Climate Research-Systems Analysis and Technology Evaluation (IEK-STE), 52428 Jülich, Germany; d.witthaut@fz-juelich.de; 2Institute for Theoretical Physics, University of Cologne, 50937 Köln, Germany; 3Department of Epileptology, University Hospital Bonn, Venusberg Campus 1, 53127 Bonn, Germany; klaus.lehnertz@ukbonn.de; 4Helmholtz-Institute for Radiation and Nuclear Physics, University of Bonn, Nussallee 14–16, 53115 Bonn, Germany; 5Interdisciplinary Center for Complex Systems, University of Bonn, Brühler Straße 7, 53175 Bonn, Germany; 6Department of Computer Science, OsloMet—Oslo Metropolitan University, P.O. Box 4 St. Olavs plass, N-0130 Oslo, Norway; pedrolin@oslomet.no

**Keywords:** stochastic processes, Kramers–Moyal equation, Kramers–Moyal coefficients, Fokker–Planck equation, arbitrary-order approximations, non-parametric estimators, Bell polynomials

## Abstract

With the aim of improving the reconstruction of stochastic evolution equations from empirical time-series data, we derive a full representation of the generator of the Kramers–Moyal operator via a power-series expansion of the exponential operator. This expansion is necessary for deriving the different terms in a stochastic differential equation. With the full representation of this operator, we are able to separate finite-time corrections of the power-series expansion of arbitrary order into terms with and without derivatives of the Kramers–Moyal coefficients. We arrive at a closed-form solution expressed through conditional moments, which can be extracted directly from time-series data with a finite sampling intervals. We provide all finite-time correction terms for parametric and non-parametric estimation of the Kramers–Moyal coefficients for discontinuous processes which can be easily implemented—employing Bell polynomials—in time-series analyses of stochastic processes. With exemplary cases of insufficiently sampled diffusion and jump-diffusion processes, we demonstrate the advantages of our arbitrary-order finite-time corrections and their impact in distinguishing diffusion and jump-diffusion processes strictly from time-series data.

## 1. Introduction

The reconstruction of stochastic evolution equations from time-series data in terms of the Langevin equation and the corresponding Fokker–Planck equation is often challenged by the inevitably finite temporal sampling of time-series data. Moreover, the Fokker–Planck equation is restricted to continuous stochastic processes, i.e., diffusion, and thus cannot adequately describe discontinuous transitions in time-series data. A more general description of continuous and discontinuous stochastic processes can be constructed using the Kramers–Moyal equation [1,2] given by
$$\frac{\partial }{\partial \tau }p\left(x,t+\tau |x^{\prime },t\right)=\sum_{n=1}^{\infty }\;{\left(\;-\frac{\partial }{\partial x}\right)}^{n}\;\;D_{n}\left(x\right)p\left(x,t+\tau |x^{\prime },t\right),$$
of which the Fokker–Planck equation is a particular case ($D_{n}\equiv 0$ for n>2). The Kramers–Moyal equation serves as a stepping stone to adequately describe time-series data with both diffusive and discontinuous characteristics, but it is nevertheless challenged by finite-time sampling in real-world data. Recent applications of the Kramers–Moyal equation include brain [3,4] and heart dynamics [5], stochastic harmonic oscillators [6], renewable-energy generation [7], solar irradiance [8], turbulence [9], nano-scale friction [10], and X-ray imaging [11,12].

Previous work has demonstrated that a finite sampling interval Δt not only influences the first- and second-order Kramers–Moyal (KM) coefficients [13] but also causes non-vanishing, higher-order (>2) coefficients [4,14,15,16,17,18,19,20]. A more recent example is the jump-diffusion process discussed in reference [3]
(1)$$dX_{t}=a\left(X_{t},t\right)dt+b\left(X_{t},t\right)dW\left(t\right)+\xi \left(X_{t},t\right)dJ\left(t\right),$$
where $a(X_{t},t)$ is a drift function, $b(X_{t},t)$ is the diffusion associated with an uncorrelated Brownian motion or Wiener process W(t), J(t) is a Poisson process with jump rate λ, independent of W(t), and $\xi (X_{t},t)$ is Gaussian-distributed N(0,s) with zero mean and variance *s*. For such jump-diffusion processes, additional influences of the finite temporal sampling need to be taken into account. As shown in reference [8], jump events produce terms of order O(Δt) in the KM coefficients of even orders and the jump rate and amplitude induce terms of order $O(\Delta t^{2})$ in all coefficients. These influences are heightened for bivariate jump-diffusion processes [21], since terms of order $O(\Delta t^{i}),i\geq 3$ impact higher-order (≥4) coefficients [22].

Although most of the aforementioned studies reported on finite-time corrections for KM coefficients and/or conditional moments of various orders, we still lack an explicit arbitrary-order correction or a closed-form solution in which the conditional moments are represented as functions of the KM coefficients and vice versa. In this article, we derive a full expansion of the generator of the Kramers–Moyal operator in exponential form for one-dimensional Markovian processes. This is equivalent to van Kampen’s system-size expansion, which is taken over a finite time interval τ [23,24]. The derivations presented henceforth are generally applicable to Markovian diffusion as well as jump-diffusion processes.

On a more general level, our solution is an explicit approximate solution of the Kramers–Moyal equation [1,2], which generalises the Fokker–Planck equation for discontinuous processes [13,23,25]. Our approximation of the Kramers–Moyal operator can be taken as an arbitrary order. In particular, we focus on the solution of this partial differential equation by representing the Kramers–Moyal operator in an exponential form and equating the conditional moments with the KM coefficients after representing the exponential operator as a power series. This representation of the exponential operator can similarly be used in other problems with an equivalent formulation [26,27,28] or similar discontinuous stochastic processes with different jump distributions, e.g., the Gamma distribution [29,30].

## 2. Mathematical Background

The Fokker–Planck(–Kolmogorov) equation (Kolmogorov forward equation or Smoluchowski equation) for the conditional probability density $p(x,t+\tau |x^{\prime },t)$, that is well-known within the fields of physics and mathematics, yields the propagation in time and space of any diffusion (thus continuous) process, is given by [31]
(2)$$\frac{\partial }{\partial \tau }p\left(x,t+\tau |x^{\prime },t\right)=\left[\frac{\partial }{\partial x}D_{1}\left(x,t\right)+\frac{\partial^{2}}{\partial x^{2}}D_{2}\left(x,t\right)\right]p\left(x,t+\tau |x^{\prime },t\right).$$We restrict our investigation to stationary processes, hence $D_{n}\left(x,t\right)=D_{n}\left(x\right)$. Equation (Equation 2) describes the evolution of, for instance, a Brownian particle (for the case $D_{1}\left(x\right)=0$), which results in the known heat equation, or more complicated Markovian motions with drift. Here, one recognises the function $D_{1}\left(x\right)$, the first KM coefficient, commonly denoted as drift, and the function $D_{2}\left(x\right)$, the second KM coefficient, commonly denoted as diffusion or volatility. The Fokker–Planck equation is, nevertheless, only valid for continuous motions and thus cannot describe jump-diffusion processes as in the case in Equation (Equation 1) or other stochastic motions with discontinuous paths.

A more general equation—the so-called Kramers–Moyal equation—takes higher-order KM coefficients $D_{n}\left(x\right),n\in N$ into account
(3)$$\frac{\partial }{\partial \tau }p\left(x,t+\tau |x^{\prime },t\right)=L_{\mathrm{KM}}p\left(x,t+\tau |x^{\prime },t\right),$$
where $L_{\mathrm{KM}}$ denotes the Kramers–Moyal operator defined as the power series [1,2]
$$L_{\mathrm{KM}}=\sum_{n=1}^{\infty }\;{\left(\;-\frac{\partial }{\partial x}\right)}^{n}\;\;D_{n}\left(x\right),$$
which we will subsequently solve for τ and an appropriate starting condition by exponentiating the Kramers–Moyal operator $L_{\mathrm{KM}}$.

When examining a stochastic process in terms of time-series data, there is no direct access to the KM coefficients $D_{n}\left(x\right)$ but rather to the conditional moments of the data. The nth-order conditional moment $M_{n}\left(x,\tau \right)$ is given by
(4)$$M_{n}\left(x^{\prime },\tau \right)=\int_{-\infty }^{\infty }{\left(x-x^{\prime }\right)}^{n}p\left(x,t+\tau |x^{\prime },t\right)dx.$$
The KM coefficients $D_{n}\left(x\right)$ can be retrieved from the conditional moments $M_{n}\left(x,\tau \right)$ via
$$D_{n}\left(x\right)=\frac{1}{n!}\lim_{\tau \to 0}\frac{M_{n}\left(x,\tau \right)}{\tau }.$$
When dealing with real-world data, we do not have access to infinite temporal resolution, meaning that the above limit τ→0 is not possible. A best-case scenario is to analyse the smallest possible temporal differences. If the data are sampled at Δt time steps, take
$$D_{n}\left(x\right)=\frac{1}{n!}\frac{1}{\Delta t}M_{n}\left(x,\Delta t\right).$$

In order to non-parametrically retrieve the conditional moments $M_{n}\left(x,\tau \right)$ from data, a set of histogram or Nadaraya–Watson estimators can be utilised (see Refs. [29,32] for details). Here, we will focus not on how to estimate the conditional moments but rather on how to derive a set of finite-time corrections to estimate the KM coefficients from conditional moments. These can be retrieved from data with software packages like kramersmoyal [33] or JumpDiff [34] in Python or Langevin [35] in R.

## 3. The Formal Solution of the Kramers–Moyal Equation and Its Approximations

First, we explicitly derive the corrective terms and subsequently link these to the results in reference [36], connecting them to the relation between statistical cumulants and moments [13].

Let us assume a well-defined initial state of the Kramers–Moyal equation be given by $\delta (x-x^{\prime })$. The formal solution of the time-dependent Kramers–Moyal equation (Equation 3) is given by
(5)$$p\left(x,t\;\;+\;\;\tau |x^{\prime },t\right)=\exp\left(\tau L_{\mathrm{KM}}\right)\delta \left(x-x^{\prime }\right)=\sum_{k=0}^{\infty }\frac{{\left(\tau L_{\mathrm{KM}}\right)}^{k}}{k!}\delta \left(x-x^{\prime }\right),$$
where $p(x,t\;\;+\;\;\tau |x^{\prime },t)$ is a normalisable function, such that $\int_{-\infty }^{\infty }p\left(x,t\;\;+\;\;\tau |x^{\prime },t\right)dx=1,\;\;\forall \left(t,\tau \right)$. We will now proceed to show the first-, second-, third-order, and arbitrary-order approximation to the solution of this partial differential equation with this particular initial condition.

### 3.1. The First- and Second-Order Approximations

The first-order approximation of the formal solution of Equation (Equation 5) is given by
$$p\left(x,t\;+\;\tau |x^{\prime },t\right)=\exp\left(\tau L_{\mathrm{KM}}\right)\delta \left(x-x^{\prime }\right)=\left[1+\tau L_{\mathrm{KM}}+O\left(\tau^{2}\right)\right]\delta \left(x-x^{\prime }\right),$$
yielding for the conditional moments $M_{n}\left(x^{\prime },\tau \right)$ in Equation (Equation 4)
$$\begin{array}{cc} M_{n}\left(x^{\prime },\tau \right)\; & ≃M_{n}^{\left[1\right]}\left(x^{\prime },\tau \right)=\int_{-\infty }^{\infty }{\left(x-x^{\prime }\right)}^{n}\left[1+\tau L_{\mathrm{KM}}\right]\delta \left(x-x^{\prime }\right)dx\\ \; & =\int_{-\infty }^{\infty }{\left(x-x^{\prime }\right)}^{n}\delta \left(x-x^{\prime }\right)dx+\tau \;\;\int_{-\infty }^{\infty }{\left(x-x^{\prime }\right)}^{n}\sum_{m=1}^{\infty }\;{\left(\;-\frac{\partial }{\partial x}\right)}^{m}\;\;D_{m}\left(x\right)\delta \left(x-x^{\prime }\right)dx\\ \; & =0+\tau \sum_{m=1}^{\infty }{\left(-1\right)}^{m}\int_{-\infty }^{\infty }D_{m}\left(x\right)\left[{\left(\;-\frac{\partial }{\partial x}\right)}^{m}\;\;{\left(x-x^{\prime }\right)}^{n}\right]\delta \left(x-x^{\prime }\right)dx\\ \; & =\tau \sum_{m=1}^{\infty }\int_{-\infty }^{\infty }D_{m}\left(x\right)\frac{n!}{(n-m)!}{\left(x-x^{\prime }\right)}^{n-m}\delta \left(x-x^{\prime }\right)dx\\ \; & =\tau \sum_{m=1}^{\infty }D_{m}\left(x^{\prime }\right)\frac{n!}{(n-m)!}\delta_{n,m}\\ \; & =\tau \left(n!\right)D_{n}\left(x^{\prime }\right), \end{array}$$
where the large square brackets indicate that the derivation operation is limited to the terms within the brackets. The superscript [1] indicates the order of approximation.

The second-order approximation is obtained in a similar fashion, now including the quadratic term from the exponential representation Equation (Equation 5), i.e.,
$$\begin{array}{cc} p(x,t\;+\;\tau |x^{\prime },t) & =\left[1+\tau L_{\mathrm{KM}}+\frac{\tau^{2}}{2}L_{\mathrm{KM}}L_{\mathrm{KM}}+O\left(\tau^{3}\right)\right]\delta \left(x-x^{\prime }\right). \end{array}$$
To alleviate the notation, we refer to the KM coefficient without explicit state dependencies, i.e., $D_{n}$. The second-order approximation $M_{n}^{\left[2\right]}\left(x^{\prime },\tau \right)$ of the n-th conditional moment in Equation (Equation 4) reads
Mn[2](x′,τ)=∫−∞∞(x−x′)n1+τLKM+τ22LKMLKMδ(x−x′)dx=Mn[1](x′,τ)+τ22∫−∞∞(x−x′)n∑p=1∞−∂∂xpDp∑m=1∞−∂∂xmDmδ(x−x′)dx=Mn[1](x′,τ)+τ22∑p,m=1∞∫−∞∞(x−x′)n−∂∂xpDp−∂∂xmDmδ(x−x′)dx=Mn[1](x′,τ)+τ22∑p,m=1∞∫−∞∞n!(n−p)!(x−x′)n−pDp−∂∂xmDmδ(x−x′)dx=Mn[1])(x′,τ)+τ22∑p,m=1∞∫−∞∞n!(n−p)!(x−x′)n−p−m(n−p)!(n−p−m)!DpDmδ(x−x′)dx+τ22∑p,m=1∞∑s=0m−1∫−∞∞n!(x−x′)n−p−s(n−p−s)!ms−∂∂xm−sDpDmδ(x−x′)dx.
The first integral is only non-vanishing if n−p−m=0 and the second integral is only non-vanishing if n−p−s=0, with s<n. Hence,
Mn[2](x′,τ)=Mn[1](x′,τ)+τ22(n!)∑m=1n−1Dn−m(x′)Dm(x′)+τ22(n!)∑s=0n−1∑m=s+1∞ms∂∂x′m−sDn−s(x′)Dm(x′).
Separating the terms between those with explicit derivatives of the KM coefficients and those without, it is immediately clear that the second-order approximation follows a structure given by the partial ordinary Bell polynomials ${\hat{B}}_{n,m}$ [37]
(6)$${\hat{B}}_{n,m}\left(x_{1},x_{2},\ldots ,x_{n-m+1}\right)=\sum \frac{m!}{j_{1}!j_{2}!\cdots j_{n-m+1}!}x_{1}^{j_{1}}x_{2}^{j_{2}}\cdots x_{n-m+1}^{j_{n-m+1}}\;.$$
where the summation is taken over $j_{1},\cdots ,j_{n-m+1}\in \left\{0,1,2,\ldots ,n-m+1\right\}$ such that
(7)$$\sum_{r=1}^{n-m+1}\;\;\;\;j_{r}=m\;\mathrm{and}\;\sum_{r=1}^{n-m+1}\;\;\;\;rj_{r}=n.$$
The first- and second-order approximations can be written with the help of the partial ordinary Bell polynomials with m=1 and m=2, respectively,
(8)$$\begin{array}{cc} M_{n}^{\left[1\right]}\left(x^{\prime },\tau \right) & =\left(n!\right)\tau {\hat{B}}_{n,1}\left(D_{1},\ldots ,D_{n}\right),\\ M_{n}^{\left[2\right]}\left(x^{\prime },\tau \right) & =\left(n!\right)\left[\tau {\hat{B}}_{n,1}\left(D_{1},\ldots ,D_{n}\right)+\frac{\tau^{2}}{2}{\hat{B}}_{n,2}\left(D_{1},\ldots ,D_{n-1}\right)+\Phi_{n}^{\left[2\right]}\right], \end{array}$$
where $\Phi_{n}^{\left[2\right]}$ incorporates all derivatives of the KM coefficients from the 2nd-order corrections, and is given by
Φn[2]=τ22∑s=0n−1∑m=s+1∞ms∂∂x′m−sDn−s(x′)Dm(x′).
To simplify the description, we introduce a short-hand notation and take the superscript (m) in the KM coefficients: $D_{p}^{\left(m\right)}\left(x^{\prime }\right)={\left(\frac{\partial }{\partial x^{\prime }}\right)}^{\;m}\;\;D_{p}\left(x^{\prime }\right)$.

These results are in line with those reported for diffusion-type processes [16,17,18,19,38], where the Kramers–Moyal operator $L_{\mathrm{KM}}=L_{\mathrm{FP}}$ reduces to the Fokker–Planck operator and we are solely left with the first two KM coefficients, as in Equation (Equation 2). In particular, applying the second-order approximation in Equation (Equation 8) to the two first KM coefficients results in
$$\begin{array}{cc} M_{1}^{\left[2\right]} & =\tau D_{1}+\frac{\tau^{2}}{2}\sum_{m=1}^{\infty }D_{m}D_{1}^{\left(m\right)},\\ M_{2}^{\left[2\right]} & =2\tau D_{2}+\tau^{2}D_{1}^{2}+\tau^{2}\left[\sum_{m=1}^{\infty }D_{m}D_{2}^{\left(m\right)}+\sum_{m=2}^{\infty }mD_{m}D_{1}^{\left(m-1\right)}\right], \end{array}$$
and truncating the sums at second order yields the expressions in reference [16].

### 3.2. The Third-Order Approximation

Before we introduce the general formalism for the arbitrary-order approximation, we explicitly derive the third-order approximation
$$p\left(x,t\;+\;\tau |x^{\prime },t\right)=\left[1+\tau L_{\mathrm{KM}}+\frac{\tau^{2}}{2}L_{\mathrm{KM}}L_{\mathrm{KM}}+\frac{\tau^{3}}{6}L_{\mathrm{KM}}L_{\mathrm{KM}}L_{\mathrm{KM}}+O\left(\tau^{4}\right)\right]\delta \left(x-x^{\prime }\right),$$
which leads to
Mn[3](x′,τ)=Mn[1](x′,τ)+Mn[2](x′,τ)+τ36∑q,p,m=1∞∫−∞∞(x−x′)n−∂∂xqDq−∂∂xpDp−∂∂xmDmδ(x−x′)dx=Mn[1](x′,τ)+Mn[2](x′,τ)+τ36∑q,p,m=1∞∫−∞∞n!(n−q−p−m)!(x−x′)n−q−p−mDqDpDmδ(x−x′)dx+τ36∑q=1n∑p,m=1∞∑s=0p∫−∞∞n!(x−x′)n−q−s(n−q−s)!ps−∂∂xp−sDq×Dp−∂∂xmDmδ(x−x′)dx=Mn[1](x′,τ)+Mn[2](x′,τ)+τ36∑q,p,m=1∞∫−∞∞n!(n−q−p−m)!(x−x′)n−q−p−mDqDpDmδ(x−x′)dx+τ36∑q=1n∑p,m=1∞∑s=0p∑k=0m∑r=0m−k∫−∞∞n!(x−x′)n−q−s−k(n−q−s−r)!psmkm−kr×−∂∂xp−s+rDq−∂∂xm−k−rDpDmδ(x−x′)dx.
Notice that the first integral is only non-vanishing for the combination q+p+m=n, which can again be expressed via the partial ordinary Bell polynomial ${\hat{B}}_{n,m}$, where m=3, for the third-order approximation. The second expression requires q+s+k=n as well as p+r≠s∧m−r≠k. Separating these again into two expressions, one with and another without derivatives, we can express the third-order approximation as
(9)$$\begin{array}{cc} M_{n}^{\left[3\right]}\left(x^{\prime },\tau \right) & =\left(n!\right)\left[\tau {\hat{B}}_{n,1}\left(D_{1},\ldots ,D_{n}\right)+\frac{\tau^{2}}{2}{\hat{B}}_{n,2}\left(D_{1},\ldots ,D_{n-1}\right)+\Phi_{n}^{\left[2\right]}\right.\\ & \;\;\;\;\;\;\;\left.+\frac{\tau^{3}}{6}{\hat{B}}_{n,3}\left(D_{1},\ldots ,D_{n-2}\right)+\Phi_{n}^{\left[3\right]}\right], \end{array}$$
where $\Phi_{n}^{\left[3\right]}$ incorporates all derivatives of the KM coefficients from the third-order corrections
(10)Φ1[3]=τ36∑p,m=1∞∑r=0mmr−∂∂xp+rD1(x)−∂∂xm−rDp(x)Dm(x).

Here, we compare our derivation to the derivation of third-order approximation in Gottschall and Peinke [16]. We note that our derivation takes the general form of the Kramers–Moyal operator, to which the Fokker–Planck operator is circumscribed. From Equation (Equation 9), we derive an identical expression for the Fokker–Planck operator reported in reference [16]. Since the Fokker–Planck operator is limited to second-order terms, i.e., $D_{n}\equiv 0$ for n≥3, the sum in Equation (Equation 10) can be express in full. For the first conditional moment $M_{1}^{\left[3\right]}$, we obtain the corrective terms ${\tilde{\Phi }}_{1}^{\left[3\right]}$ given by
Φ˜1[3]=τ36∑p,m=12∑r=0mmr−∂∂xp+rD1(x)−∂∂xm−rDp(x)Dm(x).=τ36D1(1)D1(1)D1+D1(2)D1D1+3D1(1)D1(2)D2+2D1(3)D1D2+D1(2)D2(1)D1+D1(2)D2(2)D2+2D1(3)D2(1)D2+D1(4)D1D1,
which is identical to Equation (A1) in the Appendix of reference [16]. Similarly, for the second conditional moment $M_{2}^{\left[3\right]}$, we obtain the corrective terms ${\tilde{\Phi }}_{2}^{\left[3\right]}$
Φ˜2[3]=τ36∑p,m=12∑r=0mpmr−∂∂xp+r−1D1(x)−∂∂xm−rDp(x)Dm(x)+τ36∑p,m=12∑r=0m−1mm−1r−∂∂xp+rD1(x)−∂∂xm−r−1Dp(x)Dm(x)+τ36∑p,m=12∑r=0mmr−∂∂xp+rD2(x)−∂∂xm−rDp(x)Dm(x)=τ333D1D1(1)D1+7D1(2)D1D2+4D1(1)D1(1)D2+3D1(1)D2(1)D1+4D1(1)D2(2)D2+7D1(2)D2(1)D2+4D1(3)D2D2+D2(2)D1D1+2D2(3)D2D1+D2(2)D2(1)D1+D2(2)D2(2)D2+2D2(3)D2(1)D2+D2(4)D2D2
which is in agreement with Equation (A2) in the Appendix of reference [16]. A similar derivation can be found in Appendix B of reference [8], which also yields congruent findings for the first two conditional moments of jump-diffusion processes. However, no explicit expression for all terms is given in either publication.

As a simple rule of thumb, one can confer if the result is correct, as follows: the sum of the order of the KM coefficients subtracted by the derivation operation must equal *n*, the order of the conditional moment being calculated. In the notation used in this work, the *sum of subscripts* minus *the sum of superscripts* must equal the order *n* of the coefficient under investigation.

### 3.3. Arbitrary-Order Approximation

We now derive the arbitrary-order corrections of the Kramers–Moyal operator. This is done by induction from the previous derivations, whilst disregarding any emerging terms with derivatives of the KM coefficients
$$M_{n}^{\left[m\right]}\left(x^{\prime },\tau \right)\;=\;\int_{-\infty }^{\infty }\;{\left(x-x^{\prime }\right)}^{n}\sum_{k=1}^{m}\frac{\tau^{k}}{k!}L_{\mathrm{KM}}^{k}\delta \left(x-x^{\prime }\right)dx=n!\sum_{k=1}^{m}\left[\frac{\tau^{k}}{k!}\prod_{\sigma (k,n)}^{m}\;\;D_{\sigma (k,n)}+\Phi_{n}^{\left[k\right]}\right],$$
with σ(k,n) a partition of a set of k∈N obeying Equation (Equation 7). This, in turn, is the same as a collection of partial Bell polynomials, namely
$$M_{n}^{\left[m\right]}\left(x^{\prime },\tau \right)=\left(n!\right)\sum_{k=1}^{m}\left[\frac{\tau^{k}}{k!}{\hat{B}}_{n,k}\left(D_{1},D_{2},\ldots ,D_{n-k+1}\right)+\Phi_{n}^{\left[k\right]}\right],$$
where we combine terms with derivatives in $\Phi_{n}^{\left[k\right]}$. If we disregard the derivative terms, the summation has an upper bound, namely m≤n. This is directly seen as the Bell polynomials are similarly bounded, and thus we arrive at
(11)$$M_{n}\left(x^{\prime },\tau \right)=\left(n!\right)\sum_{k=1}^{n}\frac{\tau^{k}}{k!}{\hat{B}}_{n,k}\left(D_{1},D_{2},\ldots ,D_{n-k+1}\right).$$
neglecting the derivative terms Φ.

From the perspective of estimation, the aim is to determine the KM coefficients $D_{n}\left(x^{\prime }\right)$, however what we have expressed here is the relation of the conditional moments $M_{n}\left(x^{\prime },\tau \right)$. As we now have an explicit relation in terms of partial Bell polynomials, we will invert the relation and express the KM coefficients $D_{n}\left(x^{\prime }\right)$ as functions of the conditional moments $M_{1}\left(x^{\prime },\tau \right),\ldots ,M_{n}\left(x^{\prime },\tau \right)$.

Note that the first conditional moment $M_{1}\left(x^{\prime },\tau \right)$ is solely a function of the first KM coefficient $D_{1}\left(x^{\prime }\right)$. The second conditional moment $M_{2}\left(x^{\prime },\tau \right)$ is a function of the second KM coefficient $D_{2}\left(x^{\prime }\right)$, and by substitution, a function of the first conditional moment $M_{1}\left(x^{\prime },\tau \right)$, given by Equation (Equation 11). Subsequently $M_{3}\left(x^{\prime },\tau \right)$ is a function of $D_{3}\left(x^{\prime }\right)$, $M_{2}\left(x^{\prime },\tau \right)$, and $M_{1}\left(x^{\prime },\tau \right)$. Thus, by recursively substituting the n−1 KM coefficients by their expressions via the conditional moments, we obtain a relation of $D_{n}\left(x^{\prime }\right)$ as a function of the $M_{n}\left(x^{\prime },\tau \right),M_{n-1}\left(x^{\prime },\tau \right),\ldots ,M_{1}\left(x^{\prime },\tau \right)$ conditional moments.

To this end, we rewrite Equation (Equation 11) in terms of the partial exponential Bell polynomials $B_{n,m}$
$$B_{n,m}\left(x_{1},x_{2},\ldots ,x_{n-m+1}\right)=\sum \frac{n!}{j_{1}!j_{2}!\cdots j_{n-m+1}!}{\left(\frac{x_{1}}{1!}\right)}^{j_{1}}{\left(\frac{x_{2}}{2!}\right)}^{j_{2}}\;\;\;\;\cdots {\left(\frac{x_{n-m+1}}{(n-m+1)!}\right)}^{j_{n-m+1}},$$
where the summation terms obey the constraints of the Bell polynomials given in Equation (Equation 7). This can be expressed through the partial ordinary Bell polynomials in Equation (Equation 6) as
$${\hat{B}}_{n,m}\left(x_{1},x_{2},\ldots ,x_{n-m+1}\right)=\frac{m!}{n!}B_{n,m}\left(1!\cdot x_{1},2!\cdot x_{2},\ldots ,\left(n-m+1\right)!\cdot x_{n-m+1}\right).$$
Thus, Equation (Equation 11) reads
$$M_{n}\left(x^{\prime },\tau \right)=\sum_{k=1}^{n}B_{n,k}\left(1!\tau D_{1},2!\tau D_{2},\ldots ,\left(n-k+1\right)!\tau D_{n-k+1}\right).$$
We can then utilise the reciprocal relations of the partial exponential Bell polynomials: for a set of variables $y_{1},\cdots ,y_{n}$, defined as functions of *n* other variables $x_{1},\cdots ,x_{n}$ given by
(12)$$y_{n}=\sum_{k=1}^{n}B_{n,k}\left(x_{1},x_{2},\ldots ,x_{n-k+1}\right),$$
the inverse relation holds
(13)$$x_{n}=\sum_{k=1}^{n}{\left(-1\right)}^{k-1}\left(k-1\right)!B_{n,k}\left(y_{1},y_{2},\ldots ,y_{n-k+1}\right).$$

With this, we can finally express any KM coefficients $D_{n}\left(x^{\prime }\right)$ from the nth-order power series expansion, neglecting the derivative terms Φ, as 
(14)$$D_{n}\left(x^{\prime },\tau \right)=\frac{1}{n!}\frac{1}{\tau }\sum_{k=1}^{n}{\left(-1\right)}^{k-1}\left(k-1\right)!B_{n,k}\left(M_{1},M_{2},\ldots ,M_{n-k+1}\right).$$

We note here that these relations are equivalent to the relation between cumulants and (non-central) moments of a probability distribution [13,36]. Let M(y) be the moment-generating function, such that
$$M\left(y\right)=1+\sum_{n=1}^{\infty }\frac{\mu_{n}^{\prime }y^{n}}{n!}=\exp\left[\sum_{n=1}^{\infty }\frac{\kappa_{n}y^{n}}{n!}\right]=\exp\left[K(y)\right],$$
with $\mu_{n}^{\prime }$ the (non-central) moments and K(x) the cumulant-generating function. For n<4, the cumulants $\kappa_{n}$ and the central moments are the same (e.g., the mean and variance). This is not the same for higher cumulants and moments. The relation between the cumulants $\kappa_{n}$ and the (non-central) moments $\mu_{n}^{\prime }$ is given by the reciprocal relation of the Bell polynomials, as in Equations (Equation 12) and (Equation 13). This is in line with our exponential representation of the Kramers–Moyal operator. Here, the KM coefficients are the cumulants (with the exception of the τ term).

## 4. Exemplary Cases with Constant Diffusion and Constant Jumps

Here, we present two illustrative examples: first, a constant diffusion process, the  Ornstein–Uhlenbeck process; secondly, we augment this process with jumps to obtain a jump-diffusion process. We implement the corrective terms derived thus far to show the impact of the finite-time corrections. This choice of parameters, i.e., constant diffusion and constant jumps, considerably simplifies Equation (Equation 1) to
(15)$$dX_{t}=-aX_{t}dt+bdW\left(t\right)+\xi dJ\left(t\right),$$
where $-aX_{t}$ is the state-dependent linear drift function, with a>0, also denoted mean-reverting strength, b>0 a constant diffusion, W(t) a Brownian motion or Wiener process, ξ a state-independent and normally distributed jump amplitude with zero mean and variance *s*, and J(t) a Poisson process with jump rate λ. Note that the conventional Ornstein–Uhlenbeck process is recovered if we omit the jump process.

We have derived an expression for the conditional moments $M_{n}\left(x,\tau \right)$ as a function of the KM coefficients $D_{n}\left(x\right)$, given by Equation (Equation 11), which is valid for any Markovian diffusion or jump-diffision process. For our particular application to the Poissonian jump-diffusion process in Equation (Equation 1) we require at least the first six KM coefficients/first six moments. These are given by
$$\begin{array}{cc} M_{1} & =\tau D_{1},\\ M_{2} & =2\tau D_{2}+\tau^{2}D_{1}^{2},\\ M_{3} & =6\tau D_{3}+6\tau^{2}D_{1}D_{2}+\tau^{3}D_{1}^{3},\\ M_{4} & =24\tau D_{4}+12\tau^{2}\left(2D_{1}D_{3}+D_{2}^{2}\right)+12\tau^{3}D_{1}^{2}D_{2}+\tau^{4}D_{1}^{4},\\ M_{5} & =120\tau D_{5}+120\tau^{2}\left(D_{1}D_{4}+D_{2}D_{3}\right)+60\tau^{3}\left(D_{1}^{2}D_{3}+D_{1}D_{2}^{2}\right)+20\tau^{4}D_{1}^{3}D_{2}+\tau^{5}D_{1}^{5},\\ M_{6} & =720\tau D_{6}+360\tau^{2}\left(2D_{1}D_{5}+2D_{2}D_{4}+D_{3}^{2}\right)+120\tau^{3}\left(3D_{1}^{2}D_{4}+6D_{1}D_{2}D_{3}+D_{2}^{3}\right)\\ \; & \;\;\;\;+60\tau^{4}\left(2D_{1}^{3}D_{3}+3D_{1}^{2}D_{2}^{2}\right)+30\tau^{5}D_{1}^{4}D_{2}+\tau^{6}D_{1}^{6}. \end{array}$$
We invert this expression explicitly using Equation (Equation 14) and report on the KM coefficients as functions of the conditional moments, which are given by
(16)$$\begin{array}{cc} D_{1} & =\frac{1}{1!}\lim_{\tau \to 0}\frac{1}{\tau }M_{1},\\ D_{2} & =\frac{1}{2!}\lim_{\tau \to 0}\frac{1}{\tau }\left[M_{2}-M_{1}^{2}\right],\\ D_{3} & =\frac{1}{3!}\lim_{\tau \to 0}\frac{1}{\tau }\left[M_{3}-3M_{1}M_{2}+2M_{1}^{3}\right],\\ D_{4} & =\frac{1}{4!}\lim_{\tau \to 0}\frac{1}{\tau }\left[M_{4}-4M_{1}M_{3}-3M_{2}^{2}+12M_{1}^{2}M_{2}-6M_{1}^{4}\right],\\ D_{5} & =\frac{1}{5!}\lim_{\tau \to 0}\frac{1}{\tau }\left[M_{5}-5M_{1}M_{4}-10M_{2}M_{3}+30M_{1}M_{2}^{2}+20M_{1}^{2}M_{3}\right.\\ \; & \;\;\;\left.-60M_{1}^{3}M_{2}+24M_{1}^{5}\right],\\ D_{6} & =\frac{1}{6!}\lim_{\tau \to 0}\frac{1}{\tau }\left[M_{6}-6M_{1}M_{5}-10M_{3}^{2}-15M_{2}M_{4}+30M_{2}^{3}+120M_{1}M_{2}M_{3}\right.\\ \; & \;\;\;\left.+30M_{1}^{2}M_{4}-270M_{1}^{2}M_{2}^{2}-120M_{1}^{3}M_{3}+360M_{1}^{4}M_{2}-120M_{1}^{6}\right]. \end{array}$$

We again note that these expressions are valid for any case of diffusion and jump-diffusion processes. In the first case, where there are no jump terms in Equation (Equation 15), i.e., the Ornstein–Uhlenbeck process, we know that all KM coefficients $D_{n}\left(x\right)$ with n≥3 are zero. However, this is not the case when estimating the coefficients from time-series data, i.e., from one realisation of the stochastic process sampled at finite resolution. It is common to find that these terms do not vanish due to finite-time effects. In our second case with a jump-diffusion process, the KM coefficients $D_{n}\left(x\right)$ with n≥3 can be related directly to the jump parameters. These relations were derived in reference [3], and are given by
(17)$$\begin{array}{cc} D_{1}\left(x\right) & =a(x),\\ D_{2}\left(x\right) & =\frac{1}{2}\left[b{\left(x\right)}^{2}+s\lambda \right],\\ D_{2n}\left(x\right) & =\frac{s^{n}\lambda }{2^{n}\left(n!\right)}, \end{array}$$
where $\langle \xi^{2n}\rangle =\frac{\left(2n!\right)}{2^{n}\left(n!\right)}{\langle \xi^{2}\rangle }^{n}=\frac{\left(2n!\right)}{2^{n}\left(n!\right)}s^{n}$, for Gaussian distributions with zero mean and variance *s*.

We will now compare the derived theoretical corrections to KM coefficients estimated from numerically generated time-series data. In Figure 1 and Figure 2, we display the second-, fourth-, and sixth-order KM coefficients $D_{2}\left(x\right)$, $D_{4}\left(x\right)$, and $D_{6}\left(x\right)$ estimated with the first-order, second-order, and full-order approximations given by Equation (Equation 16) (or in general Equation (Equation 14)). The full-order approximations have the same order as the KM coefficients, i.e, second-, fourth-, and sixth-order approximation for $D_{2}\left(x\right)$, $D_{4}\left(x\right)$, and $D_{6}\left(x\right)$, respectively. For the data shown in Figure 1, we use a Euler–Maruyama scheme to numerically integrate an Ornstein–Uhlenbeck process Equation (Equation 15) (without the jump terms) with parameters: drift a=1.0 and diffusion b=0.5 (λ=s=0.0). We numerically integrate this process with a coarse time-step Δt=0.1 to deliberately emphasise the finite-time effects on the aforementioned KM coefficients. For example, the second-order KM coefficients $D_{2}\left(x\right)$ takes a quadratic form, despite the fact that the diffusion term is constant. The KM coefficients $D_{4}\left(x\right)$ and $D_{6}\left(x\right)$ are not truly zero, as would be expected for purely diffusive processes [39,40], due to the finite-time effects, but the full-order finite-time correction approximates the theoretical values with far greater detail.

For the data shown in Figure 2 we follow a similar approach, now augmenting the Ornstein–Uhlenbeck process with Poissonian jumps, i.e., as given in Equation (Equation 15). The parameters are as follows: drift a=0.5, diffusion b=0.5, jump amplitude with a Gaussian distribution with variance s=0.75 and zero mean, a Poissonian jump rate λ=0.6, and a time step Δt=0.05. For this process, we know the higher-order KM coefficients $D_{4}\left(x\right)$ and $D_{6}\left(x\right)$ reflect the presence of discontinuous paths, which, for our particular case of the Poissonian jump-Ornstein–Uhlenbeck process, we know the explicit inversion in Equation (Equation 17) (cf. reference [3]). For the chosen coarse time step, we notice that the estimations do not correspond exactly with the theoretical values, regardless of the order of finite-time correction chosen. This can likely be traced back to the limitations of the Kramers–Moyal equation to fully capture discontinuous stochastic processes (cf. reference [41]). Nevertheless, the higher-order finite-time corrections approximate the theoretical values with greater accuracy.

We note here that the parameter estimation from data heavily depends on the number of data points and the sampling rate of numerically simulated or real-world time-series data. Real-world time-series data can often be sampled at higher sampling rates, but not always in such a large number of datapoints. A closer inspection of the limitations of both the sampling rate and the number of data points in parameter estimation is necessary, but falls outside the scope of this publication. Moreover, it should be emphasised that, prior to any examination of time-series data within the purview of either the Fokker–Planck or the Kramers–Moyal equation, the Markov property of the data must be account for, i.e., a vanishing memory of the increments of the data. This can be examined, for example, via the Chapman–Kolmogorov equality [13].

Summarising our findings, we conclude that our proposed arbitrary-order finite-time corrections considerably help in differentiating one-dimensional purely diffusive processes and jump-diffusion processes, as these accurately show that higher-order KM coefficients $D_{n}\left(x\right)$, n≥2 vanish for purely diffusive processes. These arbitrary-order finite-time corrections should now also be considered for *N*-dimensional stochastic processes. A first examination of the second-order finite-time corrections for two-dimensional processes was recently addressed in reference [22]. Note that the one-dimensional second-order finite-time correction for these KM coefficients was recently addressed in another publication [34]. Here, it is extended to arbitrary order.

## 5. Implementation: Symbolic Calculations in Python

In this section, we implement the main results from above to compute the moments into available software packages, e.g., kramersmoyal [33] and JumpDiff [34] in Python or Langevin [35] in R, or any self-made parametric or non-parametric estimator. In order to facilitate numerical implementations of the higher-order corrections, we include a short Python script to obtain the non-derivative corrections to any desired order, with the desired truncation of the power-series expansion.

First, we present a Python code to numerically generate the conditional moments $M_{n}\left(x^{\prime },\tau \right)$ as functions of the KM coefficients $D_{n}\left(x^{\prime }\right)$, as given in Equation (Equation 14). Here the parameter n indicates the order of the KM coefficients/moments *n* and the parameter m the order of the correction, with m ≤ n. We utilise Python’s symbolic language library sympy [42].



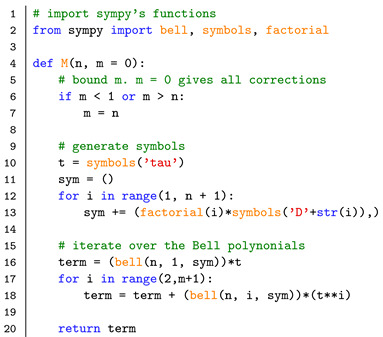



To generate the KM coefficients $D_{n}\left(x^{\prime }\right)$ as function of the conditional moments, the following must be implemented:
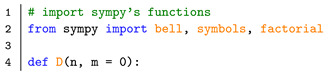

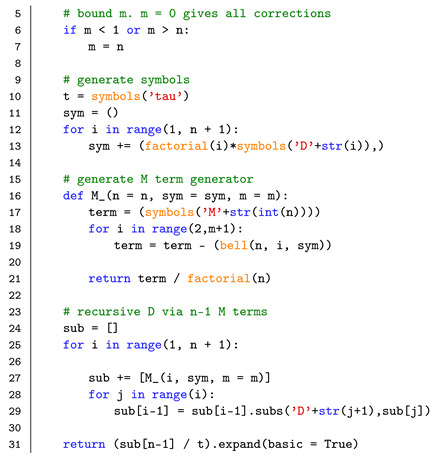


## 6. Conclusions

We have presented a set of arbitrary-order finite-time corrections to the Kramers–Moyal operator, solved by exponentiating the Kramers–Moyal operator, equivalent to van Kampen’s system-size expansion. We expressed the exponential operator as a power series and worked out each element of the series, ultimately combining it in a series representation via the partial Bell polynomials. We obtained a closed form for the set of arbitrary-order finite-time corrections relating the conditional moments to the Kramers–Moyal coefficients. Moreover, by representing the arbitrary-order finite-time corrections with partial Bell polynomials, we derived a reciprocal relation for the conditional moments and the Kramers–Moyal coefficients. This provided a closed-form representation of the Kramers–Moyal coefficients via conditional moments, which is crucial for time-series data estimation. We included two illustrative cases of poorly sampled diffusion and jump-diffusion processes with constant diffusion and constant jumps, demonstrating the suitability of our corrections for a non-parametric estimation of higher-order Kramers–Moyal coefficients. Our corrections approximated the theoretical values with a high degree of accuracy and help to distinguish processes with and without jumps. We are confident that our arbitrary-order finite-time corrections contribute to an improved reconstruction of stochastic evolution equations from empirical time-series data.

## Figures and Tables

**Figure 1 entropy-23-00517-f001:**
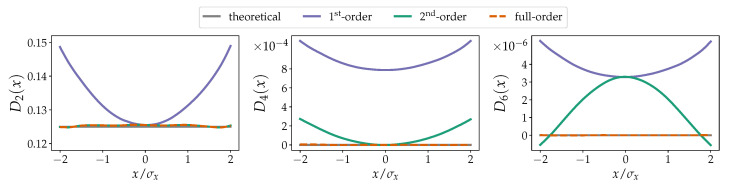
Non-parametrically estimated KM coefficients of an Ornstein–Uhlenbeck process (given by Equation (Equation 15) without the jump terms) with drift a=1.0 and diffusion b=0.5 (λ=s=0.0). The abscissas are re-scaled by the standard deviation $\sigma_{x}$ of *x*. From left to right are shown the second-, fourth-, and sixth-order KM coefficients $D_{2}\left(x\right)$, $D_{4}\left(x\right)$, and $D_{6}\left(x\right)$. There are no corrections to the drift term $D_{1}\left(x\right)$. Note that for $D_{2}\left(x\right)$ the 2nd-order and the full-order corrections are identical. The impact of solely using the second-order approximation for $D_{6}\left(x\right)$ is also evident. The coefficients $D_{4}\left(x\right)$ and $D_{6}\left(x\right)$ are theoretically zero. In all cases, the improvements to the estimation of the respective KM coefficients $D_{n}\left(x\right)$ are clear. The grey lines indicate the theoretical values. The numerical integration has a total time of $5\times 10^{5}$ and a time step Δt=0.1 ($5\times 10^{6}$ datapoints) with a Euler–Maruyama scheme [33]. Consistent results are obtained when considering a much finer numerical time step Δt of integration and subsequently down-sampling the data.

**Figure 2 entropy-23-00517-f002:**
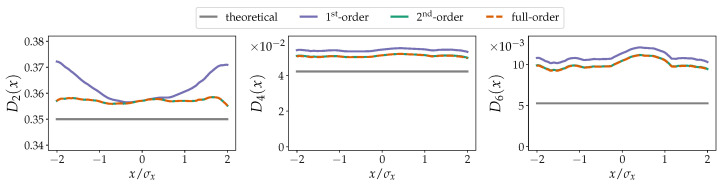
Non-parametrically estimated KM coefficients of a jump-diffusion process given by Equation (Equation 15) with drift a=0.5, diffusion b=0.5, jump amplitude s=0.75, and jump rate λ=0.6. The abscissas are re-scaled by the standard deviation $\sigma_{x}$ of *x*. From left to right are shown the second-, fourth-, and sixth-order KM coefficients $D_{2}\left(x\right)$, $D_{4}\left(x\right)$, and $D_{6}\left(x\right)$. There are no corrections to the drift term $D_{1}\left(x\right)$. Note that for $D_{2}\left(x\right)$ the 2nd-order and the full-order corrections are identical. The impact of solely using the second-order approximation for $D_{6}\left(x\right)$ is also evident. The coefficients $D_{4}\left(x\right)$ and $D_{6}\left(x\right)$ are theoretically zero. In all cases, the improvements to the estimation of the respective KM coefficients $D_{n}\left(x\right)$ are clear. The grey lines indicate the theoretical values. The numerical integration has a total time of $5\times 10^{5}$ and a time step Δt=0.1 ($5\times 10^{6}$ datapoints) with a Euler–Maruyama scheme [33].

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
