# Peer review of "Arbitrary-Order Finite-Time Corrections for the Kramers–Moyal Operator"

_entropy, 2021, doi:10.3390/e23050517_

Round 1

Reviewer 1 Report

The manuscript describes a procedure to estimate  the Kramers-Moyal coefficient using time series harvested at finite time. Pretty remarkably, the  estimator is exact, namely allows computing the finite-time corrections at any order. To the best of my knowledge, this result is novel, and therefore I strongly recommend its publication. I have a minor concern on the numerical tests presented in page 11. In these tests  the "ground truth" stochastic differential equations are integrated with a large time step. I would suggest presenting also results obtained by using a much  smaller time step,  undersampling a posteriori the time series at an arbitrarily large Dt. This, in my opinion, would allow demonstrating the quality of the estimator at an even larger Dt, without worrying about trivial integration errors induced by the Euler-Maruyama scheme.

Reviewer 2 Report

Let me first congratulate the authors with this excellent piece of work. Providing a proper relationship between conditional moments and conditional cumulants should be considered an important contribution to the field, which certainly merits publication.

Inferring dynamical characteristics from data is a timely and urgent matter, for which exploiting the Kramer-Moyal expansion forms an excellent starting point. The current manuscript will certainly boost its applicability, in particular when it comes to analyzing poorly sampled signals and/or jump-diffusion processes.

Despite my enthusiasm, however, I realized that the manuscript does deserve proper copy-editing as its lexical quality could and should be improved – next to a plenitude of typos, the syntax and semantics sometimes appears more German than English (e.g., some very long sentences are hard to digest). I strongly recommend consulting a native English academic for improving legibility, but here abstain from providing nitpicking details.

That said, I prefer to focus on several issues on the content in the hope for clarification.

1. Equations (11) and (14) are listed as “first [and second] final formula” suggesting that these forms, together will Bell’s polynomials, have first been derived by the authors. This is clearly not that case as the explicit relationships between cumulants Dk and moments Mn can be found in Prohorov & Rozanov (1969) “Probability Theory”, Springer, page 165. They can also be found in Risken (1996) “The Fokker-Planck equation”, Springer, page 18 (with reference to Prohorov & Rozanov). While the first reference is entirely absent in the manuscript, the second is listed albeit in another context (and the 4th rather than the 3rd print of that book but I trust that this equation is still in there). The only difference that I can find is the explicit inclusion of the time step τ, which is trivially resulting from the definition of Dk. That is, the authors should temper the suggestion that this is a novel result and provide proper credits and references.

Note that this does not diminish the current contribution as explicating these relationships is typically absent in current works on the Kramers-Moyal expansion as doing so is algebraically laborious. Providing symbolic coding via Python will clearly pave an easy route to implement this into the different Python modules listed in the manuscript. After all, I consider the practical benefits of this work above the theoretical novelty.

2. As for the practical use I recommend highlighting a few things more than in the current version:

- As evident from the equations and nicely illustrated in Figure 1, the ‘corrections’ do not affect the diffusion coefficient. This is, in fact, good news for the community as in the vast majority of studies only the diffusion coefficient and, of course, the drift coefficient have been reported. I.e. these results are still valid, aren’t they?

- The correction is especially important for D4 as this is typically assessed in view of Pawula’s conjecture and the subsequent reduction to mere diffusion processes. While 10-4 (in Figure 1) appears small, it certainly differs from zero, which would let me conclude that the Kramers-Moyal expansion cannot be truncated here. The full-order correction, however, seems to be several orders of magnitude smaller which seems to render truncation proper. I consider this a very important result.

- The time step in the in silico data is large, which of course stresses the problem of finite time steps way more than one would expect in current days experimental recordings. A note on this regard would be appreciated.

- The size of the time step could hence be reduced but given the research group’s interest in jump-diffusion processes one may argue that it cannot be small enough. Again, a clarifying note on this account could clearly relieve implications of the current report on studies of simple diffusion processes (i.e. comparably smooth data) and – by the same token – underscore its utter importance for less trivial cases.

- The simulation contains a substantial number of samples (5·106), which in many experimental settings is far beyond reach. Here, I am wondering how sample size will affect the difference between the different orders of correction – I can imagine that the uncertainty of the conditional moments and/or cumulants can readily exceed any difference between even 1st and 2nd order. Studying this in detail may fill another paper but a corresponding note would be much appreciated.

3. To more general issues often strike me when it comes to inferring the stochastic dynamics from data via the Kramers-Moyal expansion.

(a) The applicability of the approach crucially requires Markovianity of the underlying process. Unfortunately, this is largely ignored (for the current simulation it can of course be assumed). Yet, without proper testing via, e.g., the Chapman-Kolmogorov equality, any analysis of ‘real’ data via the Kramer-Moyal expansion will remain questionable. Again, a note (of warning) would be appreciated.

(b) Equations (11) & (14) apply to the one-dimensional case. What is the authors’ thought about extending this to the N-dimensional case? This might be a hard nut to crack, yet it does deserve discussion.

Taken together, once lexical quality has been improved and once the real novelty of the work has been clarified, I can recommend publishing the manuscript.

Reviewer 3 Report

Please see the attached report.
